# COVID-19: Neurological Considerations in Neonates and Children

**DOI:** 10.3390/children7090133

**Published:** 2020-09-10

**Authors:** Carl E. Stafstrom, Lauren L. Jantzie

**Affiliations:** 1Division of Pediatric Neurology, Departments of Neurology and Pediatrics, The Johns Hopkins University Hospital and School of Medicine, Baltimore, MD 21287, USA; 2Departments of Pediatrics, Neurosurgery, and Neurology, The Johns Hopkins University Hospital and School of Medicine, Baltimore, MD 21287, USA; LJantzie@jhmi.edu

**Keywords:** COVID-19, coronavirus, SARS-CoV-2, neonate, neurological, brain, neurotropism, cytokine storm, neuroinflammation, neurodevelopment

## Abstract

The ongoing worldwide pandemic of the novel human coronavirus SARS-CoV-2 and the ensuing disease, COVID-19, has presented enormous and unprecedented challenges for all medical specialists. However, to date, children, especially neonates, have been relatively spared from the devastating consequences of this infection. Neurologic involvement is being increasingly recognized among adults with COVID-19, who can develop sensory deficits in smell and taste, delirium, encephalopathy, headaches, strokes, and peripheral nervous system disorders. Among neonates and children, COVID-19-associated neurological manifestations have been relatively rare, yet reports involving neurologic dysfunction in this age range are increasing. As discussed in this review, pediatric neurologists and other pediatric specialists should be alert to potential neurological involvement by this virus, which might have neuroinvasive capability and carry long-term neuropsychiatric and medical consequences.

## 1. Introduction

Severe and at times fatal symptoms caused by the novel human coronavirus, Severe Acute Respiratory Syndrome (SARS)-CoV-2, and the associated coronavirus disease 2019 (COVID-19), are ravaging the world. While symptoms of COVID-19 are primarily pulmonary (fever, dry cough, fatigue, pneumonia), it is becoming increasingly recognized that multiple organ systems can be affected, including the brain, with neurological involvement affecting up to ~36% of patients [1,2,3,4,5]. Information gained from studies of related coronaviruses in recent epidemics of Severe Acute Respiratory Syndrome (SARS, 2002) and Middle East Respiratory Syndrome (MERS, 2012) suggests that all three coronaviruses might have neurologic consequences [6,7], though the relative severity and frequency of neurologic involvement caused by coronaviruses varies and thus the extent to which SARS and MERS epidemics inform our understanding of COVID-19 remains unclear [5]. Nevertheless, the possibility has been raised that SARS-CoV-2 could invade the brain and cause neurological disease [2,8]. While appealing conceptually, data supporting the idea that the SARS-CoV-2 virus can infect the peripheral and central nervous systems (PNS, CNS) are limited, as discussed below. Table 1 lists definitions of relevant terms that are often used in the literature. Neurotropic viruses vary in their invasiveness, virulence, and propensity to cause inflammation [9].

The purpose of this review is twofold: (1) to discuss the available data about COVID-19 infections in neonates and children, and (2) to provide a perspective about potential neurologic involvement in neonates and children with COVID-19 infections, in view of neurobiological development.

A few points need clarification up front. First, data about the virus and its effects are accumulating rapidly and our understanding of its consequences will evolve over time. Second, much of the literature about COVID-19 currently exists as case reports or small series; obviously, the impact of such publications is limited, and greater understanding will emerge only as large, rigorous studies are published. Third, we must be mindful that a positive test for SARS-CoV-2 in a patient with a neurological symptom does not necessarily imply that the virus caused the symptom.

## 2. COVID-19 in Neonates and Children

The COVID-19 epidemic has escalated rapidly and spread widely across the globe, with cases continuing to accrue at an alarming rate. The first case was reported from Wuhan, China in mid-December 2019 and three months later, in March 2020, the World Health Organization declared COVID-19 a pandemic. As of this writing (late August, 2020), more than 23 million cases of COVID-19 have been documented worldwide (with many more mildly symptomatic cases likely not reported), with over 800,000 deaths, and more than 175,000 deaths in the United States alone (www.cdc.gov, accessed 24 August 2020). 

Documentation of the numerous clinical presentations, manifestations, and disease course have proliferated in the medical literature. A PubMed search (23 July 2020) using the keyword COVID-19 revealed an astounding number of published reports already (34,310), over the course of only a few months. Of those citations, only 501 reports (1.5%) also included the keyword neonate, attesting to the low published incidence in newborns. When searching PubMed with the key terms COVID-19, neonate and neurological or brain, fewer than 10 articles emerged. Therefore, at least so far, COVID-19 does not seem to be affecting neonates very often from a neurological point of view; PubMed counts probably underestimate the occurrence of neurological involvement in children, being biased toward areas of the world with greater medical resources. These observations do not preclude the potential for neonatal brain involvement in COVID-19 nor exclude the possibility of long-term medical, neurodevelopmental, and psychosocial consequences of the disease. Indeed, as time goes on, a wider spectrum of neurologic manifestations will likely emerge with as yet undetermined long-term sequelae.

As the COVID-19 pandemic continues, certain trends are becoming evident. First, while the number of cases and deaths continue to rise, the disease does not affect infants and children nearly as frequently as adults [9,10]. To date, approximately 2–5% of cases of COVID-19 involve children, who appear to be less severely affected than adults, mainly with pulmonary symptoms [11,12,13]. Second, disease severity in children who develop COVID-19 is usually milder than in adults, and children with severe disease often have an underlying co-morbidity such as immunosuppression [10,14]. Indeed, there is accumulating evidence that adults with COVID-19 infection manifest with multiple organ system involvement, including the CNS and PNS, and that older and sicker individuals carry a higher risk for neurologic problems [1,4,15]. These age-dependent differences in disease expression and severity have clear implications for healthcare professionals who deal with the pediatric population because children remain at risk for incurring and spreading the virus, yet many remain asymptomatic. Several hypotheses have been posited as to why children are less affected by COVID-19, including age-related differences in immune responses [16], a neutralizing antibody response due to prior exposure to coronaviruses [17], lower prevalence of co-morbidities in children, and age-specific differences in SARS-CoV-2 receptor function [18], neurovirulence, intrinsic biological protective mechanisms, and other host factors [19]. None of these hypotheses is supported compellingly by extant data at present.

While the number of affected neonates and children remains small, pediatric practitioners cannot become complacent about the potential for neurologic involvement in COVID-19. Furthermore, the recently described multisystem inflammatory syndrome-children (MIS-C), raises the specter that COVID-19 or its after-effects also target children (see Section 4) [20,21]. As data from China, Europe and other areas affected early by the COVID-19 pandemic are reported, some patterns are emerging regarding pregnant women and neonates. First, it is clear that vertical transmission COVID-19 from a pregnant mother to her fetus occurs quite rarely. More than a dozen publications attest to this observation, together encompassing over 100 patients (Table 2 lists a few relevant publications, selected from the largest available series and omitting small series and single case reports). None of these reports documents unequivocal vertical transmission. In many of the babies, onset of symptoms occurred in the neonatal period but not immediately at birth, so the exact timing of infection remains uncertain. Overall, the data does not support robust transplacental transfer of SARS-CoV-2, but recent case reports are providing proof-of-principle that the virus can be transmitted intrauterine from infected mother to fetus [22,23]. An especially apropos case demonstrated maternal viremia, placental infection shown by immunohistochemistry, and high placental viral load with subsequent neonatal viremia, implying transplacental transfer of SARS-CoV-2 from pregnant mother to fetus [24]; this newborn presented with neurological symptoms as discussed in Section 3.

Of all the infants reported during the first month of life, most had documented exposure to affected family members [13,32] which emphasizes the importance of controlling horizontal virus transmission from affected family members to the neonate [33]. While this trend may need revision [27,34], so far, it is encouraging that most COVID-19-positive pregnant women do not transmit the disease to their unborn children. Similarly, there is no evidence that COVID-19-positive pregnant women incur COVID-19 or develop more severe disease than similar-aged women who are not pregnant. However, the impact of chronic inflammation caused by maternal viral infection on fetal development, pregnancy outcomes and long-term neurodevelopment is unknown. Similarly, the timing of maternal viral infection with respect to major milestones of in utero neurodevelopment (i.e., second trimester vs. late third trimester) is an unknown and critical consideration for further research and long-term neurodevelopmental followup. There is legitimate concern about the impact of acute and chronic stress during the pandemic (i.e., worry about the medical complications of SARS-CoV-2 infection, family disruption, job loss, economic pressures, educational uncertainties, food availability, etc.) on the pregnant patient and developing fetus [35,36,37,38].

Furthermore, it is reassuring that there is no definitive evidence that the virus is present or can be transmitted in the breast milk of COVID-19-positive women [28,39,40]. The current American Academy of Pediatrics recommendation is that COVID-19–positive mothers can breast feed directly while wearing a mask or feed expressed breast milk, using appropriate breast and hand hygiene [41]. Extensive guidelines are available regarding principles of management of pregnant women with COVID-19 and their newborns [40,41,42]. Again, all of these observations are preliminary and subject to modification over time.

## 3. Neurological Involvement in COVID-19

Although COVID-19 primarily affects the pulmonary system, it is a multisystem infection (e.g., gastrointestinal tract, kidneys, liver, heart) and involvement of the PNS and CNS are increasingly recognized [43,44,45,46]. Data on neurological signs and symptoms are limited but increasing, with a wide spectrum of acute and chronic manifestations becoming apparent [47]. In a series of 214 hospitalized adults with COVID-19, 88 of whom had “severe” infections, 36.4% of the entire group was reported to manifest some neurologic involvement, including alteration of consciousness, encephalopathy, headache, cerebrovascular disease, and skeletal muscle injury (myalgia, weakness) [1].

### 3.1. Cerebrovascular Disease

Ischemic strokes, many affecting young adults with large vessel occlusions, have garnered considerable concern that the etiology may be a prothrombotic state caused by virus-induced inflammation of the vascular epithelium [48,49]. Many of the young adult stroke victims had other vascular risk factors such as diabetes or hypertension, which emphasizes the importance of comorbidities with systemic inflammatory conditions in disease manifestations and severity. Stroke has not been reported in children with COVID-19 [50].

### 3.2. Encephalitis

The occurrence of encephalitis remains controversial, as virus has not been recovered from cerebrospinal fluid (CSF) [51] and overall, a surprisingly small number of COVID-19 patients develop classic encephalitic symptoms. Autopsy studies are beginning to be published. The wide spectrum of postmortem findings include mostly secondary changes to the CNS such as hypoxemia and ischemia, rarely localized perivascular and interstitial neuroglial activation with neuronal loss and axonal degeneration [52,53], and no other major CNS abnormalities [54]. No pediatric autopsy cases have reported neuropathological involvement. Clearly, more data are needed before this issue can be clarified.

### 3.3. Seizures and Other CNS Symptoms

In the Chinese series [1], only 2 out of 214 patients had seizures (1%), which is not greater than the general population, so it is uncertain whether infected patients are at higher risk. The few patients with seizures are reported mainly in case reports [55,56]. The lack of frequent seizures is rather curious, especially if encephalopathy is indeed a frequent complication of COVID-19; this data may be related to sampling bias rather than actual non-occurrence. A few reports of seizures have appeared in adults using electroencephalography (EEG) [57,58], but a more concerted effort to evaluate the brain electrophysiology of children with COVID-19 would be informative. The examples of seizures in children with COVID-19 described in Section 3.6 appear to be largely anecdotal. Many additional cases are necessary to conclude whether there is increased seizure susceptibility in the pediatric population.

Other CNS symptoms include headache, dizziness and delirium, all of which can occur as a nonspecific consequence of systemic infection or inflammation of the respiratory tract as well as via a CNS mechanism. Although headaches are reported frequently, the pain often appears to be nonspecific or associated with inflammation or migraine exacerbation rather than meningeal irritation [59].

### 3.4. Hypogeusia, Hyposmia

The most commonly reported symptoms related to the PNS are decreased taste (hypogeusia or ageusia) and smell (hyposmia or anosmia) [60,61]. A neural mechanism is suspected for hyposmia in COVID-19, because decreased smell is often the first symptom experienced and occurs in mild disease in the absence of significant local inflammation or mucosal congestion that are typical of the more benign coronavirus or non-coronavirus nasal infections [62]. A few adolescents with COVID-19 have been reported with decreased taste or smell [63]; these symptoms appear to be very uncommon in children but deserve a more concerted ascertainment effort.

### 3.5. Demyelinating Disorders

Several cases of Guillain-Barré syndrome (GBS) have been reported in adults with COVID-19, raising the possibility of post-infectious autoimmune responses against the PNS [64,65]. Two case reports of children with GBS who developed COVID-19 symptoms about 3 weeks later confirms that GBS can occur with COVID-19, though this association remains quite rare given the widespread prevalence of COVID-19 [66,67]. Finally, the possibility of central demyelination has been raised, e.g., in the form of multiple sclerosis (MS), among patients with COVID-19 [68]; this concern is relevant in that various disease-modifying agents used to treat MS could theoretically exacerbate MS symptoms [69]. Fortunately, there is no evidence that COVID-19 triggers central demyelinating disease in children [50]. 

The lack of unequivocal reports of SARS-CoV-2 being recovered from the CSF of individuals affected with presumed neurological involvement nor in brain tissue from the limited number of autopsied cases strengthens the possibility that the virus does not often directly cause the symptoms but rather, that the neurological sequelae are secondary to hypoxia, cytokine involvement, or some other non-direct mechanism (see Section 6). It is appropriately concerning that chronic neurologic diseases such as epilepsy, amyotrophic lateral sclerosis, multiple sclerosis, etc., might be exacerbated during concurrent COVID-19 infection or that COVID-19 may unmask preexisting CNS pathology that might have been unrecognized or asymptomatic.

### 3.6. Examples of Children with Neurological Involvement

As just discussed, neurologic involvement in children, and in particular neonates, with COVID-19 appears to be scarce but may be under reported [70]. A few selected examples of case reports of neurologic involvement in neonates and children are presented in Table 3; such case reports have limited generalizability, and many lack sufficient details to ascribe causality between SARS-CoV-2 and neurologic symptoms. Most children were assumed to have contracted COVID-19 from a family member and some children had concurrent infection with other viruses, confounding any argument for causality. Importantly, CSF was negative for SARS-CoV-2 in all children on whom spinal fluid was obtained. All children recovered within a few days or weeks, contrasting with the severe and prolonged courses in many adults. Available evidence does not allow distinction between a direct effect of SARS-CoV-2 causing neurologic dysfunction, versus the symptoms instead being secondary to an over activated immune response (see Section 5). In summary, case reports of neurological involvement in babies and children are rare but accumulating, and the recovery of most infants with early neurologic symptoms implicates some virus- or host-related factors that minimize massive neurological devastation. 

## 4. Neurological Signs and Symptoms in Multi-System Inflammatory Syndrome (MIS-C)

This newly recognized Kawasaki syndrome-like hyperinflammatory disorder presents with acute hypotension and cardiogenic shock and is proliferating across the globe. It is likely a post-infectious syndrome or inflammatory reaction following asymptomatic or mildly symptomatic COVID-19 [76]. Children can develop toxic shock-like symptoms, hypoxia-ischemia, and significant end organ damage to the heart, kidneys, and other organs. While definitive data is not available, there is concern that the inflammation that hallmarks MIS-C may have adverse consequences on the developing brain. While no consistent neurologic picture has emerged, several MIS-C patients have had CNS involvement as part of their course. In a small series of 6 children with MIS-C, 4 patients had neurologic symptoms, including headache, altered mental status, and aseptic meningitis [77]. Headache was the most common symptom in a series of 58 children with MIS-C associated with SARS-CoV-2, affecting 26% of patients [78]. A series 21 children from France notes that 57% of patients were “irritable” and another 29% had “other neurological features”, though these were not specified [79]. In a larger survey of 186 children, 5–11% had neurologic involvement, depending on age, including encephalitis, seizures or mental status alteration, but details are not provided [80]. Finally, 4 of 27 children with COVID-19 associated MIS-C developed new neurologic symptoms including encephalopathy, headache, weakness, ataxia, and dysarthria [81]; two patients had lumbar punctures and CSF was negative for SARS-CoV-2 in both. Three of the four patients had an EEG; each showed diffuse slowing. Brain MRI scans of all four children showed abnormal signal intensities of the splenium of the corpus callosum (a finding seen in previous cases of encephalopathy and thought to indicate inflammation-induced focal myelin edema [81,82]. A recent systematic review of eight studies notes neurological symptoms in ~25–50% of children with MIS-C, depending upon inclusion criteria [83]. A putative impact on the immature CNS and developing immune system, including neural-immune maturation, cannot be overlooked, and the long-term neurologic impact of both COVID-19 and MIS-C on the developing brain need urgent elucidation.

## 5. Coronavirus Infectivity

SARS-CoV-2 is a highly contagious and pathogenic RNA virus that is transmitted via droplet, contact, or aerosol routes. The virus gains entry into epithelia of the pulmonary system (upper or lower respiratory tracts), digestive tract, or nasopharynx. The virus is composed of single-stranded RNA enveloped by surface proteins (S, E, M, N). The spike (S) glycoprotein serves as the attachment site onto angiotensin converting enzyme type 2 (ACE-2) receptors on epithelial membranes. The normal function of ACE-2 receptors is to provide protection against pulmonary and endothelial injury [84]. SARS and SARS-CoV-2 share ~79% genome sequence identity and both viruses infect epithelium by the binding of spike proteins to ACE-2 receptors. While most prevalent on airway and pulmonary epithelium [85], ACE-2 receptors are also reportedly present to a lesser degree on neurons and perhaps glia [2].

Binding of the S protein to ACE-2 receptors leads to proteolytic cleavage of S by the transmembrane protease TMPRSS2 [5]. Viral RNA then enters the epithelial cell and replicates rapidly, translating viral proteins and inducing a host immune response. This immune response can be adaptive, attacking and inactivating the virus. By contrast, the immune response can be maladaptive and induce a massive immune reaction, accompanied by a hyper-inflammatory response hallmarked by excessive cytokine secretion and signal transduction (cytokine storm), and robust cellular immune activation and recruitment [86]. The large-scale cytokine storm consists of a massive release of pro-inflammatory humoral agents such as interleukin-6 (IL-6), interferon gamma (IFN-Y), MCP-1/CCL2 (monocyte chemoattractant protein 1/chemokine ligand 2), IL-1, IL-12, IL-8, TNFα (tumor necrosis factor alpha), and CXCL 10 (C-X-C motif chemokine ligand 10) that exacerbate the underlying pathophysiology [87,88]. This cytokine release subsequently feeds forward an overactive and dysregulated cellular immune response defined by macrophage, monocyte, neutrophil, and T-cell hyperactivation and recruitment [89]. The impact of this systemic cytokine storm on neurodevelopment is under investigation in preclinical models [90] and should be the focus of future prospective studies. Subsequently, the replicated viruses exit the cell, leading to further infection.

It is unknown why children, and neonates in particular, seem to be relatively resistant to COVID-19 and its severe symptoms, including neurological manifestations. The cytokine response to coronavirus infection appears to be less robust in young children although the recognition of MIS-C may suggest host-dependent genetic susceptibility to enhanced cytokine and/or inflammatory responses [91], but other mechanisms are also plausible [19]. It remains controversial whether ACE-2 inhibitors would provide symptomatic relief or prevent the COVID-19 disease, and evidence for the effectiveness of these agents in children and neonates is not yet available [92].

## 6. SARS-CoV-2 Neurologic Mechanisms

The cellular and molecular basis of SARS-CoV-2 neurotropism, neuroinvasiveness, and neurovirulence are poorly understood [9]: does the virus get into the brain and if so how, and what does it do in the CNS once there (e.g., infects neural cells? causes disease?). Neurological involvement in COVID-19 might be associated with at least four potential mechanisms: 1. A direct neurotropic or neuroinvasive effect of SARS-CoV-2 (e.g., anosmia, encephalopathy), 2. A secondary effect of the systemic inflammatory responses triggered by the viral infection (e.g., encephalopathy), 3. A secondary effect associated with the vascular and prothrombotic effect of the viral infection on the CNS or PNS vasculature (e.g., strokes, necrotizing leukoencephalopathy), 4. An immune-mediated para-infectious or post-infectious autoimmune effect in response to the viral infection (e.g., GBS, acute disseminated encephalomyelitis). Figure 1 summarizes, in schematic fashion, some hypothetical possibilities about how the virus may infect the brain directly, whether the neurological symptoms and signs may be related to systemic or hyperactivation of immune responses, or both [87]. It is important to consider the mechanisms associated with neurological manifestations of COVID-19, with an aim toward developing therapeutic options. The possibilities of direct neurotropism and hyper-responsiveness to immune activation (cytokine storm) are considered separately below, though these mechanisms might work synergistically.

Definitive demonstration of direct viral invasion would require a positive CSF reverse transcriptase-polymerase chain reaction (RT-PCR) for SARS-CoV-2, recovery of infective virus from the CSF as demonstrated by viral cultures or “plaque assay” [93], intrathecal synthesis of antibodies to SARS-CoV-2, or autopsy-demonstrated SARS-CoV-2 antigen or RNA in brain tissue [5]. Current published evidence meeting these strict criteria is minimal. While it is plausible that the virus infects the brain through one of the anatomical pathways discussed below, the lack of viral recovery from the CNS gives pause to that notion. Neuroinvasion has been demonstrated for the related SARS and MERS viruses [94], but SARS-CoV-2 has not been recovered from the CSF or brain tissue. Animal models of SARS and MERS have shown that the virus can enter through epithelium of the nasopharynx and travel retrogradely to the CNS [95,96]. Interestingly, wild type mice are not vulnerable to infection and disease by human coronaviruses, but transgenic mice with human ACE-2 receptors do develop respiratory and neurological symptoms when infected [95,97]. In such transgenic mice, intranasal exposure to SARS or MERS leads to brain infection. One of the proposed portals of entry is via olfactory sensory neurons, crossing the cribiform plate into the olfactory bulb, with subsequent retrograde travel along the olfactory nerve (cranial nerve I) to the brainstem, thalamus, and basal ganglia, all areas that are connected to the olfactory cortex. Please note that it has yet to be proven that SARS-CoV-2 infects olfactory sensory neurons. Emerging animal models may clarify whether SARS-CoV-2 is similarly neuroinvasive as SARS and whether this isage dependent [97,98]. However, since mice are not naturally susceptible to the clinical and immunopathological manifestations of coronaviruses affecting humans, translational studies of pathogenic mechanisms and vaccine development become complicated. Extensive efforts to modify mice with transgenic approaches have begun to provide informative models. 

As mentioned, the olfactory epithelium has been touted as a potential site of viral entry into the brain, and hence explain hyposmia [99].Detailed genetic and immunohistochemical examinations of cell types of the olfactory system reveal that ACE-2 and TMPRSS2 are present on olfactory epithelial cells (especially supporting or “sustenacular” cells) but not on olfactory sensory neurons themselves [54,85,100]. Moreover, there is some evidence of virus-induced cell death in other coronavirus infections but not yet for SARS-CoV-2 [84,101].

Likewise, the virus might enter via the sensory system of the tongue that mediates taste, with transmission via cranial nerves VII, IX, and X to the nucleus tractus solitarius, thalamus and eventually, brain. Finally, trigeminal nociceptors via cranial nerve V from either the corneal epithelium or buccal epithelium could theoretically reach the CNS. These potential pathways could explain the symptoms of hypogeusia and altered vision. However, SARS-CoV-2 has not been recovered from the brain. Transynaptic transport from lower respiratory tract mechano- and chemoreceptors to the brainstem medullary cardiorespiratory centers has been proposed as a hypothetical mechanism that could exacerbate brainstem dysfunction and perhaps even worsen respiratory effort [102]; however, this hypothesis lacks objective validation and remains controversial.

Other potential routes for virus to enter the CNS are through the bloodstream (hematogenous) or via disruption of the blood brain barrier (BBB). From the systemic circulation, the virus might travel to the cerebral circulation where it can damage capillary epithelium and access the brain. Interestingly, there is scant evidence that SARS-CoV-2 produces a significant or sustained viremia [103]. The BBB is essential for transport of molecules into the brain and exclusion of pathogens and overall maintenance of cerebral homeostasis [104]. The BBB is a dynamic structure, consisting of several cell types and proteins, each with its own maturational profile–astrocyte foot processes, pericytes, tight junction proteins, and extracellular matrix, providing structural and functional support. Virus attachment to ACE-2 receptors at the BBB might facilitate trafficking of the virus into the CNS, facilitating endothelial damage and edema [89]. Notably, while the BBB is structurally complete at birth and is sufficiently functional in the neonate to provide protection against many pathogens, its full physiological maturation may take several months [105]. In the context of COVID-19 infection, the BBB may be dysfunctional, disrupted either by inflammatory response or the virus itself, allowing transmission of the virus or activated immune cells from the circulation into the CNS [8,84]. The release of inflammatory cytokines by activated glia and neural mast cells exacerbate the inflammation [89]. Similarly, flow of the virus through lymphatic channels of the interstitial space of the brain could breach the blood-CSF barrier and permit virus entry [106]. To date, there is no evidence for the presence of the virus in pathological specimens of the PNS or CNS, in part due to the dearth of comprehensive autopsies [52,53,54]. Obviously, patient care has focused on critical pulmonary and life-support measures so neuropathologically-focused autopsy studies have been uncommon.

Animal studies of COVID-19 will be crucial to complement information gained from prior studies of the other coronaviruses. Such animal models will provide more information about mechanisms of virus entry into the nervous system and how the virus affects neural function, neural-immune maturation and neurodevelopment, as well as the critical and yet unanswered question of long-term neurological sequelae of COVID-19 [107]. That is, if there is predilection of the virus for certain neural structures or chronic neuroinflammation, long-term consequences may arise in various neural functions such as learning, memory, cognition, seizure predisposition, and other functions. All of this is speculative at present. Another essential question, alluded to above, is whether CNS disease contributes to the respiratory failure seen in COVID-19 patients. Ongoing or severe hypoxia can exacerbate ongoing symptoms in other organs. In particular, CNS respiratory control centers in the brain stem, nearby the vagus nerve, has been speculated to play a role in respiratory failure [102,108].

## 7. Conclusions—A Cautionary Tale

At present, there is some reason for guarded optimism for young patients within the devastating COVID-19 pandemic. Children, particularly neonates, are less likely to become infected and develop severe symptoms, and their propensity to spread the virus is controversial [109]. There is at best slight evidence for vertical transmission of SARS-CoV-2 or COVID-19 disease from pregnant mother to fetus; rather, neonates are more likely incur the disease by exposure to affected individuals postnatally, and breast milk transmission has not been shown (Table 4). A variety of practical guidelines have been developed for the care of pregnant women who have or are suspected to have COVID-19 positivity. Analogous guidelines for the care of adult COVID-19 patients with neurologic problems are also available and need to be developed for children [110]. 

Regarding neurologic involvement in COVID-19, there are plausible mechanisms by which the virus can gain entry into the CNS and subsequently incur acute neurologic symptoms, either directly or through immune dysfunction (Table 5). The occurrence of long-term medical and neuropsychiatric sequelae is unknown. Children can be resilient and yet remain vulnerable to coping with the challenges of COVID-19 in the context of other acute and chronic diseases. Youngsters may not understand the need for social distancing, prolonged quarantine, and other preventative measures, and it is anticipated that stress-related post-traumatic symptoms will develop in some young people, whether or not they actually acquire symptoms. In children with comorbid chronic conditions and developmental disabilities, the challenges are even more profound. Therefore, neuropsychological surveillance and studies of the long-lasting effects of this pandemic on neurodevelopment are critical [111]. Finally, the emergence of the hyper-inflammatory multisystem syndrome (MIS-C) supplants any conclusion that COVID-19 is benign or negligible in the pediatric age range. Therefore, it behooves neurologists and other pediatric specialists who deal with neonates and young children to be aware of the potential neurologic involvement of this novel, potentially devastating virus. Future animal models should evaluate the impact of SARS-CoV-2 on maternal infection, inflammatory signal transduction through the maternal–placental–fetal axis, and brain development. The importance of large-scale immunization should a vaccine become available, cannot be over emphasized as should the role of systemic inflammation, neuroinflammation, and neural-immune interactions in novel pathophysiology and symptomology. Additionally, mechanism-specific targeted therapies could emerge from basic science studies of SARS-CoV-2 infection. 

## Figures and Tables

**Figure 1 children-07-00133-f001:**
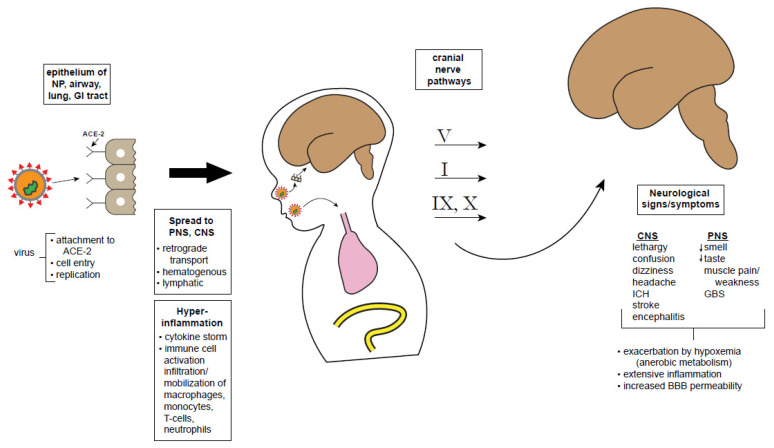
Schematic showing possible CNS entry points and effects of SARS-CoV-2. The SARS-CoV-2 virus attaches to olfactory epithelium using the ACE-2 receptor. After cell entry, the virus replicates and induces a massive immune response leading to excessive cytokine release, comprising a maladaptive immune response. Theoretically, virus particles may reach the CNS retrogradely via cranial nerve pathways: V from corneal epithelium or oropharyngeal cutaneous sensory receptors; I via the cribiform plate, infecting olfactory sensory neurons; VII and IX from tongue chemoreceptors; X via pulmonary mechanoreceptors. Once reaching CNS nuclei including brainstem and cortex, a variety of neurologic signs and symptoms are possible. However, it must be noted that the virus has not been recovered from CSF or brain tissue, making all of these pathways hypothetical at this point. Abbreviations: NP, nasopharynx; GI, gastrointestinal; ACE-2, angiotensin converting enzyme type 2 receptor; PNS, peripheral nervous system; CNS, central nervous system; ICH, intracranial hemorrhage; GBS, Guillain-Barre syndrome; BBB, blood-brain barrier.

**Table 1 children-07-00133-t001:** Definitions relevant to viral infections of the peripheral and central nervous systems.

Neuroinvasive	Virus is capable of accessing and entering the nervous system
Neurotropic	Virus is capable of infecting nerve cells once in the nervous system
Neurovirulent	Neurotropic virus is capable of causing disease in the nervous system
Neuroinflammatory	Virus causes secondary inflammatory response within the nervous system

**Table 2 children-07-00133-t002:** Selected reports evaluating potential vertical transmission of SARS-CoV-2.

Publication	Newborns (n)	SARS-CoV-2 Positive (n)	Comments
Chen et al. 2020 [25]	9	0	All C-sections
Zhang et al. 2020 [26]	16	0	All C-sections; same institution as Chen et al. 2020 [25]
Zeng et al. 2020 [27]	33	3	Tested 2-3 days postpartum; two full terms and one 31-wga premature infant; all developed pneumonia but recovered by ~1-2 weeks of life
Liu et al. 2020 [28]	19	0	PCR negative on body fluids * None developed symptoms
Schwartz et al. 2020 [29]	38	0	“No evidence that SARS-CoV-2 undergoes intrauterine or transplacental transmission from infected pregnant women to their fetuses.”
Lu & Shi 2020 [30]	-	3	3 cases mentioned; diagnosed 2-17 DOL. Details sparse
Salvatore et al. 2020 [31]	120 (DOL 1) 79 (DOL 5-7) 72 (DOL 14)	0 0 0	Cohort of infants born to SARS-CoV-2 mothers, followed through 2 weeks of life

Abbreviations: C-sections, Caesarean sections; PCR, polymerase chain reaction; wga, weeks gestational age; DOL, day of life * nasopharyngeal fluid, urine, feces, breast milk, amniotic fluid.

**Table 3 children-07-00133-t003:** Selected case reports of COVID-19 neurologic involvement in neonates and children.

Neonates (n)	Age	Presenting Symptoms	SARS-CoV-2 Testing	SARS-CoV-2 CSF	Reference
5	<3 y	Hypotonia, drowsiness	NP +	4/4 -	[71]
1	26 d	Upward eye deviation, stiffening	NP + Blood, cEEG, HUS -	NR	[72]
1	1 d	Lethargy, encephalopathy	NP +, rectal +	-	[73]
1	6 w	Upward eye deviation, leg stiffening	NP +	-	[74]
1	11 y	Seizure	NP +	NR	[75]
1	3 d	Irritability, hypertonia	Brain MRI + (transient gliosis of periventricular white matter and subcortical structures)	-	[24]

Abbreviation: y, years; d, days; w, weeks; NP, nasopharyngeal; cEEG, continuous electroencephalography; HUS, head ultrasound; MRI, magnetic resonance imaging; NR, not reported; +, positive; -, negative.

**Table 4 children-07-00133-t004:** Summary of COVID-19 infections in children.

Severe infection caused by the novel coronavirus, SARS-Cov-2, has predominant pulmonary involvement but can also affect multiple other organ systems, including the CNS and PNS.
Symptoms are less frequent and usually less severe in children and particularly in neonates.
Vertical transmission of SARS-CoV-2 from pregnant mother to fetus is rare but anecdotal case reports support this possibility.
Most cases of COVID-19 in early life are due to exposures to infected patients (horizontal transmission).
There is no reported transmission of SARS-CoV-2 via breast milk.

**Table 5 children-07-00133-t005:** Neurological involvement in COVID-19.

Acute neurological involvement in adults with COVID-19 can include decrease taste/smell, headache, confusion, peripheral nerve dysfunction, strokes, and encephalopathy.
Neurological involvement of COVID-19 in neonates and children is still quite rare but recent case reports warrant vigilant surveillance.
Neurological involvement of COVID-19 in neonates and children is still quite rare but recent case reports warrant vigilant surveillance.
SARS-CoV-2 has not been recovered from CSF or brain samples.

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
