# Peer review of "COVID-19: Neurological Considerations in Neonates and Children"

_children, 2020, doi:10.3390/children7090133_

Round 1

Reviewer 1 Report

General comments:

a) The reviewer would recommend that the manuscript undergoes proofreading and language editing for becoming slightly more concise.

b) I think this 'Review' manuscript could have been more straightforward wrt. its discussion on pediatric neurology and COVID-19. There is such a plethora of articles written on COVID-19 that some of the knowledge presented here tends to become encyclopedic.

c) I think this manuscript needs restructuring of its 'narrative'. 

d) I think the paper misses some important discussion on diseases like Guillain-Barré syndrome and Multiple Sclerosis, which are important outcomes; see, for example, previous discussion in similar MDPI journals or other journals:

  1. http://www.neurology.org/cgi/pmidlookup?view=long&pmid=32487714
  2. https://www.mdpi.com/2076-3425/10/6/345
  3. https://nn.neurology.org/content/7/5/e781.long
  4. https://link.springer.com/article/10.1007/s12035-017-0530-6

Specific comments:

Line 18: Please, highlight the limitations of case reports as the lowest part of the hierarchy of evidence”.

Line 32-34: Please, provide references to support your claims.

Line 37: Please, provide references to support your epidemiological claims and predictions expressed.

Line 41: Please, define or provide examples of these “clear implications”.

Line 43: Please, split into a new paragraph.

Line 45, 46: Could it be also that children have lower levels of ACE-2 receptors? Or, that children have stronger immunity against other cross-reacting non-SARS-1, non-MERS coronaviruses?

Line 68-69: Please, appropriately rephrase as what is reported in PubMed could be only indicative of true situations, especially in low-income countries.

Line 77-81: Please, provide at least one reference from a well-respected journal to support your claims.

Line 109: Please, could you provide a more comprehensive approach? How were these studies selected to be presented?

Line 164-196: Please, could you preferably separate in separate paragraphs?

Line 180-181: Please “tone down” these speculations.

Line 216-218: This paragraph appears wrongly misplaced in the middle of the manuscript.

Line 316: Please, provide reference to support this claim.  

Passim: Please make clear distinctions between neurotropism, neurovirulence, neuroinvasion, and neuroinflammation throughout the text.

Line 325: Please, introduce a comma after “predisposition”.  

Line 337: Please, capitalize only the letter “T” and highlight in bold all the word “Table 2”.

Line 355: Likewise, on the word “Table 3” (as comment immediately above)

Line 355: Please, avoid double-spacing.

Line 355, 356: Please, enter space after the full-stops.

Line 369: Please, avoid double-space.

Minor comments:

Line 47: Please, add a coma before “and”

Line 85: Please, highlight in bold the phrase TABLE 1, and capitalize only the letter “T”

Line 86: Please, simply phrase as “in many cases”

Line 89: “some recent case reports are emerging that provide proof-of-principle” is a rather wordy phrase, and in general, reluctance on case reports as providing proof-of-principle in our evidence-based medicine era should be taken and discussed.

Line 91: Please, do not refer to pediatrics patients as cases in our patient-centric era but rather as pediatric/ neonatal patients with COVID-19.

Line 94: Highlight in bold the phrase section 3, and capitalize S

Line 96: Make sure the syntax is correct so that the issue and not the patient emphasize, etc.

Line 97, 100: Please, introduce a hyphen

Line 103: Please, convert to i.e., (use a comma)

Line 110, 111: Please, provide appropriate capitalizations.

Line 113: “A final encouraging observation is that”. This is a wordy phrase.

Line 122: Define PNS as Peripheral Nervous System for the non-expert reader.

Line 159: Please, capitalize the diseases in question.

Line 164, 165: Please, avoid using semicolon twice.

Line 170: Please, rephrase to blood tests, not blood work.

Line 192: Is a word missing after transient?

Line 211-212: Please, avoid using twice the semicolons.

Line 213: Please, introduce a space after ‘callosum’.

Line 214-215: Please, introduce a space after the full-stop.

Line 251: Please, capitalize only the first letter of FIGURE 1, and highlight this phrase in bold.

Line 267: Underscoring is not frequently used in academic journals (e.g., versus educational notes for universities).

Line 305: Considering removing double space between “consisting” and “of”

Author Response

Author responses in BOLD.

Reviewer #1.

  1. a) The reviewer would recommend that the manuscript undergoes proofreading and language editing for becoming slightly more concise.
  2. b) I think this 'Review' manuscript could have been more straightforward wrt. its discussion on pediatric neurology and COVID-19. There is such a plethora of articles written on COVID-19 that some of the knowledge presented here tends to become encyclopedic.
  3. c) I think this manuscript needs restructuring of its 'narrative'. 

Regarding these first three points: thank you to the reviewer for suggesting that the language of this article as well as a narrative be restructured and made more clear. We agree and have done this with extensive revisions. We have reorganized much of the text, added new citations as suggested to support our contentions, and simplified some of the syntax to make the review more readable. Some excess prose (e.g., wordy descriptions of neonates, Section 3) are now summarized in (new) Table 3.

As the reviewer points out, the plethora of articles on COVID-19, especially with regard to accumulating neurologic aspects, are increasing in an almost exponential manner. Therefore, even during the couple of months during which this paper was written, new material was appearing almost daily. We agree that a lot of the reviews being published about the neurological aspects of CVID-19 are repetitive and redundant, drawing upon the same few primary articles. The most important aspect of our manuscript is that it is the first description of the neurologic aspects and neuromechanisms of COVID-19 infection in neonates and children, which differentiates this article from all of the other reviews on neurologic aspects.

Here, we attempt to provide perspective, without attempting to review the entire field, which is more the province of adult neurology. In terms of restructuring the narrative, we have reorganized the material and made it slightly more concise (as suggested). The narrative of this review proceeds from a description of the disease and its incidence, then neurological features in adults with reference to children if data is available. We then discuss the virus itself and whether/how it may be neuroinvasive, then detail some of the possible mechanisms by which the virus might affect the nervous system. We hope that these changes will satisfy the reviewer’s suggestion to “reorganize the narrative.”

  1. d) I think the paper misses some important discussion on diseases like Guillain-Barré syndrome and Multiple Sclerosis, which are important outcomes; see, for example, previous discussion in similar MDPI journals or other journals:
  1. http://www.neurology.org/cgi/pmidlookup?view=long&pmid=32487714
  2. https://www.mdpi.com/2076-3425/10/6/345
  3. https://nn.neurology.org/content/7/5/e781.long
  4. https://link.springer.com/article/10.1007/s12035-017-0530-6

We do mention Guillain-Barré syndrome in the original text and Figure 1, but elaborate on this a bit more in this revision, per the reviewer’s request, in new Section 3.5. Again, our review is pediatric oriented, and there are few pediatric case reports or series addressing Guillain-Barré and none addressing multiple sclerosis. GBS and MS are extensively covered in other COVID-19 reviews that focus on adults; these studies are cited in our paper and we included a couple suggested above by the reviewer.

Specific comments:

We thank the reviewer for reading the manuscript carefully and providing suggestions for conceptual and grammatical clarifications on a line by line basis. All errors are now corrected. Please note that line numbers often do not match exactly the original manuscript, since so much material has been removed, repositioned, or added.

Line 18: Please, highlight the limitations of case reports as the lowest part of the hierarchy of evidence”.

Excellent point. In the revision (lines 53-55) we make explicit note of the fact that most COVID-19 neurologic series involve case reports or small series, and only recently have larger series been forthcoming. We have attempted to keep up with the literature and cite it when appropriate. We completely agree that case reports are the lowest on the hierarchy of evidence and now make this point explicitly in the revision.

Line 32-34: Please, provide references to support your claims.

References 5, 6, 7 support the claims.

Line 37: Please, provide references to support your epidemiological claims and predictions expressed.

Prediction that cases will continue to rise – omitted. (They probably will increase but no reference can support such a prediction.)

Claims about disease severity: refs 9, 10.

Line 41: Please, define or provide examples of these “clear implications”.

Now lines 88-90: reworded to say that the age-specific differences in disease severity have “clear implications” because…children remain at risk, can be asymptomatic and spread the virus.

Line 43: Please, split into a new paragraph.

Now lines 90-95: We do not feel it is necessary to split this paragraph and in fact, splitting will disrupt the flow of the discussion. The theme of the paragraph is age-dependent differences, and we feel that the paragraph flows smoothly as is.

Line 45, 46: Could it be also that children have lower levels of ACE-2 receptors? Or, that children have stronger immunity against other cross-reacting non-SARS-1, non-MERS coronaviruses?

Now lines 90-97: Thank you for the ideas; these are now incorporated with new references.

Line 68-69: Please, appropriately rephrase as what is reported in PubMed could be only indicative of true situations, especially in low-income countries.

Now lines 71-74: Excellent point. As discussed above, the literature is proliferating so rapidly that relevant med citations are increasing on a day by day basis and may largely reflect the bias of resource-rich areas of the world from which academic articles are more likely to be forthcoming.

Line 77-81: Please, provide at least one reference from a well-respected journal to support your claims.

Now lines 123-125. We already have references cited from prestigious journals (#20 and 21 are from Lancet). In our expanded and more concise discussion of MIS-C (Section 4) there are references from Brit Med J, JAMA, NEJM.

Line 109: Please, could you provide a more comprehensive approach? How were these studies selected to be presented?

Now lines 161 et seq.: Studies selected for old Table 1 (now Table 2) were the largest series available, omitting single case reports. That is now stated in the text.

Line 164-196: Please, could you preferably separate in separate paragraphs?

Now lines 244-285 plus new Table 3: We decided to shorten the text by putting that material into table form (new Table 3).

Line 180-181: Please “tone down” these speculations.

This material has been deleted.

Line 216-218: This paragraph appears wrongly misplaced in the middle of the manuscript.

Now lines 319-329: Actually, this paragraph is not wrongly placed. It is the introduction to the biology of the virus itself and serves as a precursor to discussions of neural virulence and mechanism. Therefore, we would prefer to leave this verbiage as is.

Line 316: Please, provide reference to support this claim.  

Now line 432: blood-brain barrier dysfunction is discussed in references 8, 84, both now cited.

Passim: Please make clear distinctions between neurotropism, neurovirulence, neuroinvasion, and neuroinflammation throughout the text.

Excellent point. These terms are used almost interchangeably by many published reviews and need to be used carefully and precisely. We have now defined each and hopefully clarified their importance, in the newly added Table 1.

Line 325: Please, introduce a comma after “predisposition”.  

Done.

Line 337: Please, capitalize only the letter “T” and highlight in bold all the word “Table 2”.

Done.

Line 355: Likewise, on the word “Table 3” (as comment immediately above)

Done.

Line 355: Please, avoid double-spacing.

Do you mean extra spaces? (The text is single spaced throughout.) All extra spaces between words are now corrected. Many of those occurred artifactually as a result of converting to PDF format.

Line 355, 356: Please, enter space after the full-stops.

Done.

Line 369: Please, avoid double-space.

Done. 

Minor comments:

Line 47: Please, add a coma before “and”

Done.

Line 85: Please, highlight in bold the phrase TABLE 1, and capitalize only the letter “T”

Done.

Line 86: Please, simply phrase as “in many cases”

Done.

Line 89: “some recent case reports are emerging that provide proof-of-principle” is a rather wordy phrase, and in general, reluctance on case reports as providing proof-of-principle in our evidence-based medicine era should be taken and discussed.

We agree. The wordy phrase (new lines 139-140) has been clarified and simplified. The caution about relying on case reports was already discussed (new lines 53-55).

Line 91: Please, do not refer to pediatrics patients as cases in our patient-centric era but rather as pediatric/ neonatal patients with COVID-19.

Excellent point! Corrections made throughout article. In many instances we retain the phrase “case report” however.

Line 94: Highlight in bold the phrase section 3, and capitalize S

Done.

Line 96: Make sure the syntax is correct so that the issue and not the patient emphasize, etc.

Reworded: which emphasizes…

Line 97, 100: Please, introduce a hyphen

Done.

Line 103: Please, convert to i.e., (use a comma)

Done.

Line 110, 111: Please, provide appropriate capitalizations.

All capitalization is appropriate as is. For example, PCR is polymerase chain reaction (not a proper noun).

Line 113: “A final encouraging observation is that”. This is a wordy phrase.

Done.

Line 122: Define PNS as Peripheral Nervous System for the non-expert reader.

Done. (Also please note that a phrase such as peripheral nervous system would not be capitalized – it is not a proper noun.)

Line 159: Please, capitalize the diseases in question.

Disease names are not proper nouns. Capitalizing the disease names is not appropriate.

Line 164, 165: Please, avoid using semicolon twice.

Done.

Line 170: Please, rephrase to blood tests, not blood work.

Done.

Line 192: Is a word missing after transient?

No, it was an inadvertent extra space, now fixed.

Line 211-212: Please, avoid using twice the semicolons.

Done.

Line 213: Please, introduce a space after ‘callosum’.

Done.

Line 214-215: Please, introduce a space after the full-stop.

Done.

Line 251: Please, capitalize only the first letter of FIGURE 1, and highlight this phrase in bold.

Done.

Line 267: Underscoring is not frequently used in academic journals (e.g., versus educational notes for universities).

Agree, changed.

Line 305: Considering removing double space between “consisting” and “of”

Fixed.

Reviewer 2 Report

This is an extraordinarily well researched and referenced update on the neurological manifestations of COVID-19 in children, and helps to separate facts from unknowns and also provides interesting hypotheses leading to a cautionary tale.  That is, pediatric COVID-19 appears less severe than adults but there are many unknowns including direct versus potential inflammatory-mediated neurological effects.  

It is especially helpful to state unequivocal observations when possible, such as the lack of reported transmission of SARS-CoV-2 via breast milk and that virions have not been recovered from CSF or brain samples.  Involvement of the sustenacular cells of the olfactory epithelium is quite interesting as well as indirect evidence supporting neurogenic as opposed to local peripheral anosmia.

Please clarify what is meant by anticipating post-traumatic symptoms in some young people in the final paragraph.  Does this refer to psychological distress? 

The epidemiology is unfortunately a fast moving target and the authors may wish to update the incidence numbers to mid-August 2020, increasing the case count from 5+ to 20+ million, and deaths from >600K to >750K.

Very well done submission and contribution to the literature.

Author Response

Reviewer #2.

This is an extraordinarily well researched and referenced update on the neurological manifestations of COVID-19 in children, and helps to separate facts from unknowns and also provides interesting hypotheses leading to a cautionary tale.  That is, pediatric COVID-19 appears less severe than adults but there are many unknowns including direct versus potential inflammatory-mediated neurological effects.  

Thank you very much for the kind comments. We agree that our contribution is unique among the (now dozens) of COVID-19 neurologically oriented reviews.

It is especially helpful to state unequivocal observations when possible, such as the lack of reported transmission of SARS-CoV-2 via breast milk and that virions have not been recovered from CSF or brain samples.  Involvement of the sustenacular cells of the olfactory epithelium is quite interesting as well as indirect evidence supporting neurogenic as opposed to local peripheral anosmia.

Thank you for noting these points. They represent key observations that we hope will emerge from this review.

Please clarify what is meant by anticipating post-traumatic symptoms in some young people in the final paragraph.  Does this refer to psychological distress? 

Yes, we are referring to post-COVID psychological distress and elaborate on this point a bit further.

The epidemiology is unfortunately a fast moving target and the authors may wish to update the incidence numbers to mid-August 2020, increasing the case count from 5+ to 20+ million, and deaths from >600K to >750K.

We agree. Indeed, as discussed in the responses to reviewer #1, the unfortunate continuation and even increasing emergence of cases has occurred even during the writing of this review. We have updated our numbers as of this revision.

Very well done submission and contribution to the literature.

Round 2

Reviewer 1 Report

The reviewer is satisfied with the revisions made.